# Different Kinetics of HBV-DNA and HBsAg in HCV Coinfected Patients during DAAs Therapy

**DOI:** 10.3390/jcm11051406

**Published:** 2022-03-04

**Authors:** Piero Colombatto, Elena Palmisano, Gabriele Ricco, Daniela Cavallone, Filippo Oliveri, Barbara Coco, Antonio Salvati, Veronica Romagnoli, Lidia Surace, Marialinda Vatteroni, Mauro Pistello, Agostino Virdis, Ferruccio Bonino, Maurizia Rossana Brunetto

**Affiliations:** 1Hepatology Unit, Laboratory of Molecular Genetics and Pathology of Hepatitis Viruses, Reference Centre of the Tuscany Region for Chronic Liver Disease and Cancer, University Hospital of Pisa, 56124 Pisa, Italy; riccogabriele@gmail.com (G.R.); danielacavallone@hotmail.com (D.C.); f.oliveri@ao-pisa.toscana.it (F.O.); b.coco@ao-pisa.toscana.it (B.C.); salvatiantonio@hotmail.com (A.S.); v.romagnoli@ao-pisa.toscana.it (V.R.); lidia.surace@gmail.com (L.S.); 2Internal Medicine Unit, University of Pisa, 56126 Pisa, Italy; e.palmisano1@studenti.unipi.it (E.P.); agostino.virdis@med.unipi.it (A.V.); 3Virology Unit, University Hospital of Pisa, 56124 Pisa, Italy; l.vatteroni@ao-pisa.toscana.it (M.V.); mauro.pistello@med.unipi.it (M.P.); 4Institute of Biostructure and Bioimaging, National Research Council, 80145 Naples, Italy; ferruccio.bonino@unipi.it; 5Internal Medicine, Department of Clinical and Experimental Medicine, University of Pisa, 56126 Pisa, Italy

**Keywords:** hepatitis B virus, hepatitis C virus, co-infection, direct acting antivirals, chronic hepatitis B, HBeAg negative infection, reactivation, hepatitis B surface antigen, kinetics, interferon γ-induced protein 10

## Abstract

Direct-acting antivirals (DAAs) for hepatitis C virus (HCV) may induce hepatitis B virus (HBV) reactivations in co-infected patients, whose dynamics and outcomes could depend on the phase of HBV infection. We investigated HBsAg and HBV-DNA kinetics in fifteen untreated HBeAg Negative Infection (ENI) (4F-11M, 62.1y) and eight Nucleos(t)ide Analogs (NAs) treated Chronic Hepatitis B (CHB) (3F-6M, 54.8y) with HCV co-infection, receiving DAAs-regimens including Sofosbuvir (13) or not (10). All achieved a sustained virologic response (SVR) and normalized alanine-aminotransferase (ALT). At the direct acting antivirals’ (DAAs) baseline (BL), the HBV-DNA was undetectable (<6 IU/mL) in eight ENI and all CHB, the mean Log-HBsAg was lower in ENI than CHB (0.88 vs. 2.42, *p* = 0.035). During DAAs, HBV-DNA increased in untreated ENI by >1 Log in five and became detectable in two. Accordingly, mean BL Log-HBV-DNA (0.89) increased at week-4 (1.78; *p* = 0.100) and at the end of therapy (1.57; *p* = 0.104). Mean Log-HBsAg decreased at week-4 in ENI (from 0.88 to 0.55; *p* = 0.020) and CHB (from 2.42 to 2.15; *p* = 0.015). After DAAs, the HBsAg returned to pre-treatment levels in CHB, but not in ENI (six cleared HBsAg). Female gender and SOF were associated with a greater HBsAg decline. In conclusion, HBV reactivations during DAAs in HCV co-infected ENI caused moderate increases of HBV-DNA without ALT elevations. The concomitant HBsAg decline, although significant, did not modify individual pre-treatment profiles.

## 1. Introduction

Hepatitis B virus (HBV) and hepatitis C virus (HCV) are relevant public health infections able to cause chronic hepatitis with progressive liver diseases [1]. Dual HBV/HCV infection is not rare, since these viruses share in part the modes of transmission, and co-infection is often associated with a more severe course of the disease, as well as with a higher risk of developing hepatocellular carcinoma (HCC) [2].

In most cases of HBV-HCV co-infection, HCV appears able to inhibit HBV replication and HBsAg production by immune-mediated mechanisms, rather than through direct interactions [2,3,4]. Among the potential mechanisms of interaction, several in-vitro evidences suggest that HCV can lower HBV gene transcription and expression, although the exact mechanisms are not yet completely understood [5,6]. In addition, HCV affects innate and adaptive immune responses through its influence on the expression of interferon (IFN) regulated genes, kinase pathway, and microRNA regulation [7,8,9]. Most of these mechanisms have been hypothesized to play a role in the HBV reactivation process, once HCV infection is cleared.

The availability of the new direct-acting antivirals (DAAs) for chronic HCV infection dramatically improved the sustained virologic response (SVR) rate (>95%), and allowed for treatment of more patients with comorbidities and advanced liver damage, but at the same time entailed a subsequent risk of HBV reactivation during the course of the therapy or following HCV eradication [10,11,12]. Reactivation of HBV infection in this setting is more common in individuals with detectable serum HBsAg, and rare in patients with isolated anti-HBc antibodies. Once reactivation occurs, hepatic damage may be absent or very severe, occasionally being associated with fulminant hepatic failure leading to liver transplantation or death [13,14,15].

The greatest risk of clinically relevant HBV reactivation is associated with higher pre-treatment HBsAg levels and cirrhosis. On the other hand, DAA-treated HBV-HCV co-infected patients have significantly higher rates of HBsAg clearance, particularly those with low pre-treatment HBV-DNA and HBsAg levels [16,17]. Current HCV guidelines from the European Union, the United States, and Japan, recommend that all patients be assessed for their HBV exposure by HBsAg, hepatitis B surface and core antigen antibodies when initiating DAA therapy. Nucleos(t)ide Analogs (NAs) therapy is mandatory for cirrhotic patients, irrespective of baseline HBV DNA levels [18,19,20].

Despite the several studies that assessed the risk of HBV reactivation [10,11,12,13,14,15,16,17,18,19,20], the detailed kinetics of HBV-DNA and HBsAg during and after DAA therapy remains unknown. In particular, the demonstration of dynamic changes in the HBsAg levels after initiation of DAAs, with an initial decline and a subsequent rise, leaves some questions unanswered [17,21]. It is not clear whether NAs treatment and/or the phase of HBV infection might have an impact on the fluctuations of HBsAg. For this reason, we investigated HBsAg and HBV-DNA kinetics across DAAs therapy in fifteen HBsAg positive individuals with HBeAg Negative Infection (ENI) and in eight HBeAg negative Chronic Hepatitis B (CHB) pts receiving NAs. We found that in ENI the trend to increase of HBV-DNA occurred as early as at four weeks of DAAs therapy, did not cause clinically relevant events, but transiently affected the pre-existing kinetics of HBsAg decline. By contrast, in virally suppressed CHB, the even greater HBsAg decline observed during DAAs was progressively lost after treatment, and did not modify their pre-treatment profile.

## 2. Materials and Methods

### 2.1. Patients

Among 2235 HCV pts treated with DAAs at the Hepatology Unit of the University Hospital of Pisa (Italy) from January 2015, we identified 23 HBsAg positive/HBeAg negative patients (pts) who completed the 48-week follow-up and had enough measures of HBsAg and HBV-DNA serum levels to investigate their kinetics. This study did not require specific approval from the Ethical Committee, as HCV treatment and HBV surveillance was part of the routine clinical practice. All research was performed in accordance with the guidelines of the Declaration of Helsinki, and informed consent was obtained from all participants.

According to EASL Guidelines (GL) for HBV diagnosis and treatment [18], 15 subjects (4 females and 11 males, median age: 62.1 years) were classified as having HBeAg negative infections (ENI) with HBV-DNA persistently below 2000 IU/mL, whereas 8 (3 females and 6 males, median age: 54.8 years) were CHB pts already receiving NA treatment (entecavir: 6; tenofovir: 3).

### 2.2. Treatment Regimen

For HCV treatment, the patients received DAAs with sofosbuvir (SOF: 13 pts) or without (NO-SOF: 10 pts) for 8 weeks (4 pts treated with glecaprevir + pibrentasvir); 12 weeks (12 pts: 1 sofosbuvir + ribavirin, 1 sofosbuvir + simeprevir + ribavirin, 1 sofosbuvir + daclatasvir, 1 sofosbuvir + ledipasvir, 2 grazoprevir + elbasvir, 3 sofosbuvir + velpatasvir and 3 glecaprevir + pibrentasvir); or 24 weeks (7 pts: 2 sofosbuvir + daclatasvir, 2 sofosbuvir + daclatasvir + ribavirin, 2 sofosbuvir + ledipasvir and 1 ombitasvir + paritaprevir-ritonavir + ribavirin) due to concomitant HCV infection (genotype 1a: 4; 1b: 7; 2a/c: 3; 3a: 7; 4d: 2). The stage of liver disease was assessed prior treatment by laboratory liver function parameters, US scan and Liver Stiffness (LS) using Fibroscan^®^ (Echosense, Italy).

### 2.3. Assays

The quantification of serum HCV-RNA (COBAS^®^ AmpliPrep/COBAS^®^ TaqMan^®^ HCV Test, v2.0; dynamic range 15–1.7 × 10^6^ IU/mL; Roche Diagnostic Systems Inc., Mannheim, Germany) and HBV-DNA (COBAS^®^ AmpliPrep/COBAS^®^ TaqMan^®^ HBV Test, v2.0; dynamic range 6–1.10 × 10^8^ IU/mL; Roche Diagnostic Systems Inc., Mannheim, Germany) were performed at the Virology Unit of the University Hospital of Pisa. Quantification of serum HBsAg (Architect^®^ HBsAg assay, dynamic range 0.05–250.0 IU/mL, WHO standard; Abbott Laboratories, Chicago, IL, USA) was performed, after appropriate dilution when >250 IU/mL, at the Laboratory of Molecular Genetics and Pathology of Hepatitis Viruses of the Hepatology Unit of the University Hospital of Pisa. The measures were obtained at the following time points: 12 weeks before DAAs start (−12 w), baseline (BL), week 4 (4 w) and at the End of Therapy (EOT), and weeks 12, 24 and 48 during post-treatment follow-up (FU12, FU24, FU48). Liver function tests, including AST, ALT, albumin, total bilirubin, and INR were measured by routine biochemistry assays (Cobas Analyzer, Roche Diagnostic Systems Inc., Mannheim, Germany) at −12 w and at FU12 and FU48, whereas AST and ALT were measured at all time points. To samples with HBV-DNA below the Lower Limit of Quantification (LLQ: 6 IU/mL), the conventional values of 5 IU/mL or 1 IU/mL were attributed if HBV-DNA was qualitatively detectable yet or not. Similarly, the conventional values of 0.01 IU/mL was attributed to the samples with HBsAg levels below the LLQ (0.05 IU/mL). Reactivation of HBV infection was defined according to previous studies if HBV-DNA increased >1 Log or became detectable >100 IU/mL [17].

Interferon-gamma-induced protein 10 (IP-10) was tested at BL, 4 w, EOT and FU12 in a subgroup of 12 pts with available serum samples collected and stored at −20 °C. IP-10 serum levels were quantified by IP-10 ELISA kit (ThermoFisher, Waltham, MA, USA). The sensitivity of the test was 2.0 pg/mL, and the quantification range was 7.8–500 pg/mL.

### 2.4. Statistics

The results were expressed using median and range values or mean and standard deviation (SD) when appropriate. The logarithmic transformation was used for representing quantitative data without normal distribution. Differences between groups were analyzed using Chi-square test for categorical variables and Mann–Whitney U Test for continuous variables. Student’s *t*-test was used to compare means of paired data, and Pearson’s *r* test for correlations. We considered *p* values of <0.05 as statistically significant. Statistical analyses were performed using SPSS software (version 20.0; SPSS Inc., Chicago, IL, USA).

## 3. Results

### 3.1. Clinical and Virological Features

The main clinical and virological characteristics of the study cohort are reported in Table 1. The fifteen patients with ENI tended to be older than the eight patients with CHB (median age: 62.1 vs. 54.8 years, *p* = 0.093), but no significant differences were observed in median baseline LS (10.5 vs. 15.7 kPa; *p* = 0.495), ALT (51 vs. 44 U/L; *p* = 0.495) and HCV-RNA (6.2 vs. 6.0 Log IU/mL, *p* = 0.561). As expected, median HBsAg levels were lower in ENI than in CHB (6.61 vs. 213.19 IU/mL; *p* = 0.028). In the ENI group, HBV-DNA was undetectable in 5/15 (33.3%) subjects, and <10 IU/mL in the other 3 (20.0%); in the remaining 7 (46.7%) median HBV-DNA was 31 IU/mL (range: 10–599 IU/mL). In CHB patients, all with on-going NAs treatment, HBV-DNA was undetectable at the baseline of DAAs therapy and remained so thereafter. All patients achieved a sustained virological response (SVR) for HCV and normalized ALT at week 4 during therapy, and showed a minimal increase of the mean values from week 4 to EOT (22.90 ± 12.46 vs. 29.33 ± 19.96; *p* = 0.6951), with only one ENI patient who had elevated ALT (89 U/L) at EOT (Figure 1).

### 3.2. HBV-DNA and HBsAg Kinetics

#### 3.2.1. HBeAg Negative Infection

During DAAs, a reactivation of HBV infection occurred in 7/15 (46.7%) subjects with ENI, as HBV-DNA increased >1 Log or became detectable >100 IU/mL in 5 (33.3%) and 2 (13.3%), respectively. Overall, mean Log HBV-DNA levels were stable before DAAs (−12 w: 0.89 ± 1.00 Log, BL: 0.89 ± 0.91 Log; *p* = 0.997), and then showed a trend for higher values at 4 w (1.78 Log; *p* = 0.100) and at EOT (1.57 Log; *p* = 0.104), as compared to BL (Figure 1*)*. By contrast, mean Log HBsAg levels significantly decreased from −12 w to BL (1.13 ± 1.72 to 0.88 ± 1.78 Log; *p* = 0.002), and then continued to decline during DAAs at 4 w (0.55 ± 1.69 Log; *p* = 0.020), but rebounded at EOT (0.79 ± 1.76 Log; *p* = 0.544) to levels not significantly different from those measured at BL (Figure 1).

After DAAs, HBV-DNA returned to levels not significantly different from BL, whereas mean Log HBsAg continued progressively to decrease at FU12 (0.33 ± 2.01 Log; *p* = 0.011), FU24 (0.13 ± 1.93 Log; *p* = 0.001) and FU48 (−0.16 ± 1.86 Log; *p* = 0.001). By the end of the follow-up 6 (40.0%) patients cleared HBsAg (Figure 1).

In the 10 patients with measurable HBV-DNA levels during DAAs therapy, a significant inverse correlation (Pearson’s Correlation Coefficient R = 0.483; *p* = 0.031) was found between the maximum Log HBV-DNA increase that occurred in the treatment period and the maximum Log decline of HBsAg observed in the same period (Figure 2).

#### 3.2.2. HBeAg Negative Chronic Hepatitis B

All CHB patients were already on treatment with NA before starting DAAs (from 45–123 months) and HBV-DNA was undetectable at −12 w and at BL. None of the patients discontinued NA therapy during the study and HBV-DNA remained undetectable in all of them during and after therapy with DAAs.

Mean Log HBsAg levels were stable before DAAs (−12 w: 2.36 ± 0.88 Log, BL: 2.42 ± 1.00 Log; *p* = 0.564), significantly decreased during DAAs at 4 w (2.15 ± 0.98 Log; *p* = 0.015) and continued to decrease at EOT (2.08 ± 1.06 Log; *p* = 0.009), as compared to BL.

After DAAs, mean Log HBsAg increased progressively to levels not significantly different from BL at FU12 (2.26 ± 1.01 Log; *p* = 0.142), FU24 (2.45 ± 1.03 Log; *p* = 0.423) and FU48 (2.42 ± 1.07 Log; *p* = 0.217). By the end of the follow-up no patients cleared HBsAg. The kinetics of HBV-DNA and HBsAg in CHB is shown in Figure 3.

#### 3.2.3. Factors Influencing HBsAg Decline during DAAs

The maximum HBsAg decline observed during DAAs therapy was computable in 22 patients, as one ENI with very low level HBsAg at the screening (0.10 IU/mL) tested negative at baseline. Among categorical variables associated with a greater HBsAg decline during DAAs, there was the female gender (median: 0.48 vs. 0.27; *p* = 0.042) and the presence of sofosbuvir in the treatment schedule (median: 0.36 vs. 0.22; *p* = 0.045), but not the HBV infection phase and the fibrosis stage (Table 2).

In seven patients the HBsAg decline was greater than 0.4 Log, this event was significantly more frequent in female than in male patients (66.7% vs. 18.7%; *p* = 0.032), and showed a trend for the presence of sofosbuvir in the DAAs schedule (46.2% vs. 11.1%; *p* = 0.083).

Overall, HBsAg declined during DAAs therapy in 13/13 (100%) patients treated with SOF and in 7/9 (77.8%) patients treated with NO SOF regimens, the two patients in whom HBsAg increased were classified as ENI and did not receive NA therapy. HBsAg declined in all eight NAs treated CHB and in all six female patients, regardless of the DAAs regimens. By contrast, 2/16 (12.5%) of the males had increased levels of HBsAg during DAAs therapy; they were both ENI without NAs therapy and received NO SOF regimens. The maximum Log HBsAg decline, calculated by the difference of Log HBsAg at BL minus the minimum Log HBsAg measured in each patient during DAAs therapy, is shown (Figure 4) according to the DAAs treatment regimen, gender, and HBV antiviral therapy.

#### 3.2.4. Interferon Gamma Inducible Protein 10 during DAAs

The kinetics of the interferon gamma (IFN-γ) inducible protein of 10 kDa (IP-10) was investigated in seven ENI and in five CHB who had stored sera available at BL, week 4, EOT and at FU week 12. Mean IP-10 levels significantly decreased from BL to week 4 during DAAs therapy (2.35 ± 0.30 Log pg/mL to 1.66 ± 0.57 Log pg/mL; *p* < 0.001; *t*-test for paired data) and remained stable thereafter (Table 3). The mean IP-10 Log decline was not significantly different between ENI and CHB at week 4 (0.62 ± 0.40 vs. 0.77 ± 0.50; *p* = 0.599), EOT (0.63 ± 0.3 vs. 0.54 ± 0.43; *p* = 0.678) and at FU 12 week (0.54 ± 0.49 vs. 0.59 ± 0.5; *p* = 0.889). Log HCV-RNA decline at week 4 was significantly correlated with Log IP-10 decline at the end of follow up (R = 0.836; *p* = 0.001). Log IP-10 decline at any time point did not correlate with Log HBsAg decline and with ALT decline.

## 4. Discussion

This study investigates for the first time the combined kinetics of HBV-DNA and HBsAg during and after DAAs therapy in chronic hepatitis C patients with concomitant HBeAg Negative Infection or HBeAg negative Chronic Hepatitis B on Nucleos(t)ide Analogs.

In most cases of HBV co-infection, HCV appears to inhibit HBV replication [3,22]. Accordingly, clearance of HCV infection, either by IFN-based treatments or DAAs, can be associated with HBV reactivations followed by hepatitis flares [14,15,16,17]. The evidence that in some cases such events become clinically relevant emphasizes the need for screening, monitoring and treating HBV infection with NAs to prevent HBV reactivation during anti-HCV treatments [18,19,20]. Although the extent of the hepatic damage may reach hepatic failure and death in cirrhotic patients, in the majority of the cases only a transient mild increase of HBV-DNA occurs, and it has been suggested that it may lead to an improvement of the immune control with higher chances of HBsAg seroclearance [17]. Such evidence supports the conclusion that the individual level of immune control of HBV infection (i.e., the phase of HBV infection) is a major determinant of the risk of HBV reactivation in untreated carriers, while the fibrosis stage may influence the severity of liver damage [10,11,12,13,14,15].

To shed more light on the complex virological and immunological events driving HBV reactivations, we analyzed the relationship between the kinetics of HBV-DNA and HBsAg across DAAs therapy in the sera of fifteen untreated ENI and eight NAs-treated CHB patients. In a subgroup, we also investigated the kinetics of the IFN-γ-inducible protein of 10 kDa (IP-10), a marker of the magnitude of endogenous IFN response also known as CXCL10, which reflects the HCV-related activation of the interferon system [23].

As expected, a temporary mild increase of HBV replication occurred early during DAAs therapy in patients with untreated HbeAg negative infection (ENI), which caused an average 0.9 Log increase of HBV-DNA levels as early as at week 4 (Figure 1). Such increase was not accompanied by elevations of the ALT, which declined along with HCV-RNA. In CHB patients, HBV-DNA remained undetectable throughout DAAs treatment because of the concomitant NA therapy.

HBsAg levels showed a significant decline at week 4 in both ENI and CHB patients, followed by a rebound at EOT in ENI only, reaching levels not significantly different from those measured at BL. In CHB patients, instead, no rebound was observed at EOT, but a slower progressive increase of HBsAg levels occurring after EOT (Figure 3). Interestingly, the maximum HBsAg decline achieved during DAAs in untreated ENI was inversely related to the maximum increase in HBV-DNA (Figure 2). In other words, the greater the increase in HBV-DNA, the smaller the decline in HBsAg. This evidence is consistent with the fact that HBsAg levels at EOT were higher than at week 4 in ENI, but not in CHB patients because Nas treatment avoided HBV reactivations. Nevertheless, once DAA treatment ended and HCV was cleared, HBsAg serum levels returned to decline in ENI, and six (40%) subjects cleared HBsAg within 48 weeks from the end of DAA therapy. In co-infected patients with CHB the NA therapy was continued, but, despite the absence of HBV reactivations, HBsAg levels progressively returned to the BL levels and remained stable until the end of follow-up.

Noteworthy, the different kinetics of HBsAg between ENI and CHB were already evident before DAAs therapy. In fact, HBsAg levels in CHB patients were higher and stable, compared to ENI, in whom a progressive decline was present. Overall, our findings suggest that the treatment with DAAs caused only a transient reduction of HBsAg levels, whereas the long-term kinetics remained mainly determined by the HBV infection phase, showing a trend toward spontaneous decline in ENI. This is in agreement with previous longitudinal studies in low-viremic subjects who display a progressive HBsAg decline and very low chance of transitioning from inactive carriers to active CHB [24,25]. Indeed, it appears that DAAs treatment is unable to induce a transition from one HBV phase to another.

Among factors that could have influenced the extent of the transient HBsAg decline observed during DAAs (Table 2), we found that maximum HBsAg decline was not significantly different according to HBV infection phase and liver fibrosis stage, whereas a significantly higher decline was present in females than in males (median: 0.48 vs. 0.27; *p* = 0.042) and in patients treated with SOF based regimens than in those without SOF (median: 0.36 vs. 0.22; *p* = 0.045). Accordingly, HBsAg decreased during DAAs therapy more than 0.4 Log in 4/6 females but only in 3/22 males (*p* = 0.032), and in 6/13 patients treated with SOF but only in 1/9 of those treated without SOF (*p* = 0.083). Despite the statistically significant differences observed, we must recognize that the analysis of the factors influencing HBsAg decline in this cohort has several limitations. First of all, the small number of cases and the variety of anti-HCV treatment schedules used did not allow us to run a multivariate analysis aimed at identifying the factors independently associated with the decline. Moreover, ENIs were not treated with NAs, thus 7/15 patients experienced a reactivation of HBV replication. The increase of HBV-DNA inversely correlated with the decline of HBsAg, therefore it is likely that HBV reactivations limited the maximum HBsAg decline in the ENI group, introducing a bias in the analysis.

Nevertheless, some observations are in line with previous studies. In fact, the female gender has frequently been associated with a more benign course of HBV infection and disease [26,27,28,29]. Other authors reported an unexplained HBsAg decline during SOF therapy in HBV-inactive carriers co-infected with HCV [17,30], and the hypothesis that SOF may have some antiviral activity towards HBV has been investigated in a proof-of-concept phase-2 study in which 21 HBV mono-infected patients were enrolled. The recently posted results show a mean decline of HBsAg of 0.40 Log during a 12 week course of sofosbuvir in combination with ledipasvir, and of 0.19 Log with sofosbuvir alone, and no decline with ledipasvir alone [31]. While there are no known mechanisms that could support a direct effect of sofosbuvir and/or ledipasvir in HBsAg decline, there is no clear alternative virological explanation for the opposite kinetics of HBV-DNA and HBsAg during treatment [17,21].

It has been shown that the rapid HCV clearance achieved by DAAs therapy leads to the rebalancing of the innate antiviral responses in both the peripheral blood and liver. The consequent improvement of the natural killer cell functions might cause the restoration of the innate immunity with beneficial effects on the immune control of HBV [3]. In this study we investigated the kinetics of interferon gamma (IFN-γ) inducible protein of 10 kDa (IP-10) in seven ENI and in five CHB, who had stored sera available at BL, week 4, EOT and at week 12 FU. We found that IP-10 levels correlated to the rapid decline of HCV-RNA at week 4 (Table 3). Accordingly, the increase of HBV-DNA in our untreated ENI can be explained by the fading of HCV-induced IFN responses, and the decline of HBsAg could be the consequence of the improved natural killer cell functions. The rebound of HBsAg levels at EOT observed in ENI, instead, appears consistent with a transient expansion of HBV infection, eventually controlled by the anti-HBV immune response. In fact, after the HBV reactivation and the clearance of HCV, HBsAg levels continued to decline in ENI, but not in CHB. This different behavior suggests that the improvement of the innate immunity is only transient and does not clearly modify the outcome of HBV infection.

The asymmetry of HBV markers during DAAs, however, deserves further investigations. Most recent in-vitro studies clearly showed that co-infected cells produce fewer HBV transcripts, progeny viruses, and antigens, due to the hampered HBV replication and transcription caused by the HCV-induced IFN response [6]. The role of the adaptive immunity appears to be minor, since HCV has not been shown to affect the T cell response of HBV in coinfected patients [32], nor did HCV clearance after DAAs [33]. Taken together, our results are in line with the in-vitro evidence that HBV reactivation is the consequence of a diminished hepatic IFN response following HCV clearance. In keeping with this interpretation, however, when HCV-RNA drops at week 4 both HBV-DNA and HBsAg levels should increase. To explain the rapid HBsAg decline observed in concomitance with the increase of HBV-DNA, we can only make some speculations. One hypothesis is that NK cells, which participate in repressing HBV replication through IFN-α mediated effects with a non-specific cytotoxic phenotype, develop a more effective antiviral phenotype with increased IFN-γ production that inhibits HBsAg production more efficiently than HBV-DNA replication [34,35].

The possibility that other factors, including SOF, could have inhibited more efficiently the transcription/translation of HBsAg cannot be ruled out. Sofosbuvir, a uridine analogue active on the HCV NS5B RNA-dependent RNA polymerase (RdRp), showed a modest and transient antiviral efficacy also in patients with chronic Hepatitis E Virus (HEV), another single-stranded RNA virus that replicates by RdRp [36]. However, in vitro studies have clearly shown that sofosbuvir does not inhibit human DNA polymerases alpha, beta, and gamma at the highest concentration tested (100 μM), nor does it affect Pol II-catalyzed RNA synthesis [37,38], excluding the possibility that SOF could have affected the transcription of the mRNAs coding for HBsAg.

## 5. Conclusions

HBV reactivations during DAAs in HCV co-infected ENI caused moderate increases of HBV-DNA without ALT elevations. The concomitant HBsAg decline, although significant, did not modify individual pre-treatment profiles. The reasons for the asymmetry of HBV-DNA and HBsAg kinetics observed early during DAAs treatment remains unclear and deserves further investigations.

## Figures and Tables

**Figure 1 jcm-11-01406-f001:**
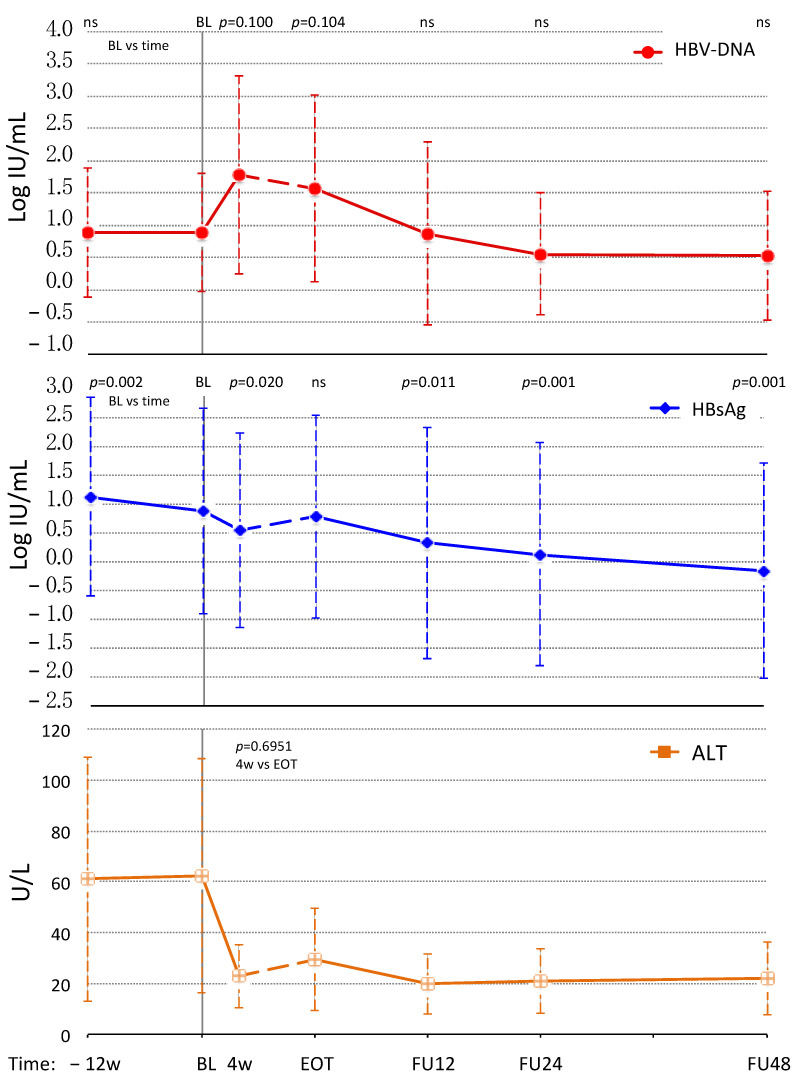
Kinetics of HBV-DNA, HBsAg and ALT across Direct Acting Antivirals (DAAs) treatment is reported for the 15 patients with concomitant HBeAg Negative Infection (ENI). None of these patients received antiviral therapy for HBV. Data are represented as mean values (points) with Standard Deviation (dashed lines). The baseline (BL) values were compared to the other time points (−12 w: 12 weeks before BL; 4 w: week 4 of DAAs therapy; EOT: End of Therapy; FU12, FU24 and FU48: Follow Up week 12, 24 and 48) using the Student’s *t*-test for means of paired data (*p* values of the most relevant comparisons are showed on the top of each graph; ns: not significant).

**Figure 2 jcm-11-01406-f002:**
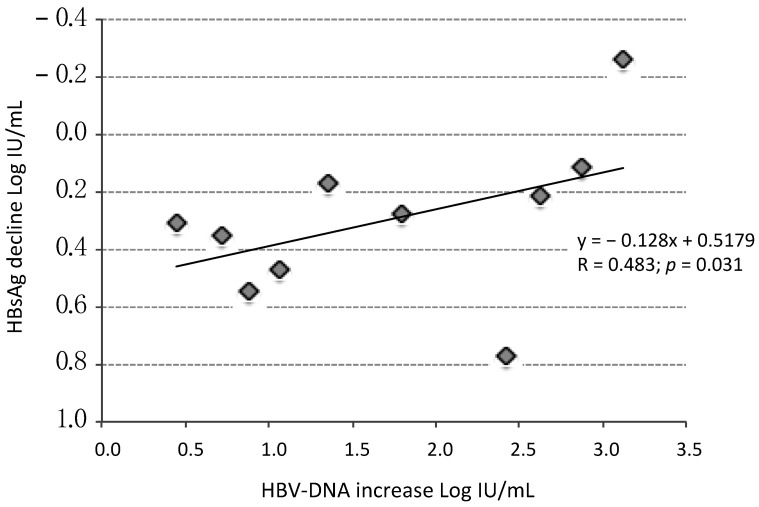
Correlation between maximum HBsAg decline during DAAs therapy (◊) and maximum increase of HBV-DNA in the same period. Higher increase of Log HBV-DNA correlates with lower decline of Log HBsAg in 10/15 patients with HBeAg Negative Infection and measurable HBV-DNA levels; in 5 patients without measurable HBV-DNA levels during therapy the maximum HBV-DNA increase was not computable.

**Figure 3 jcm-11-01406-f003:**
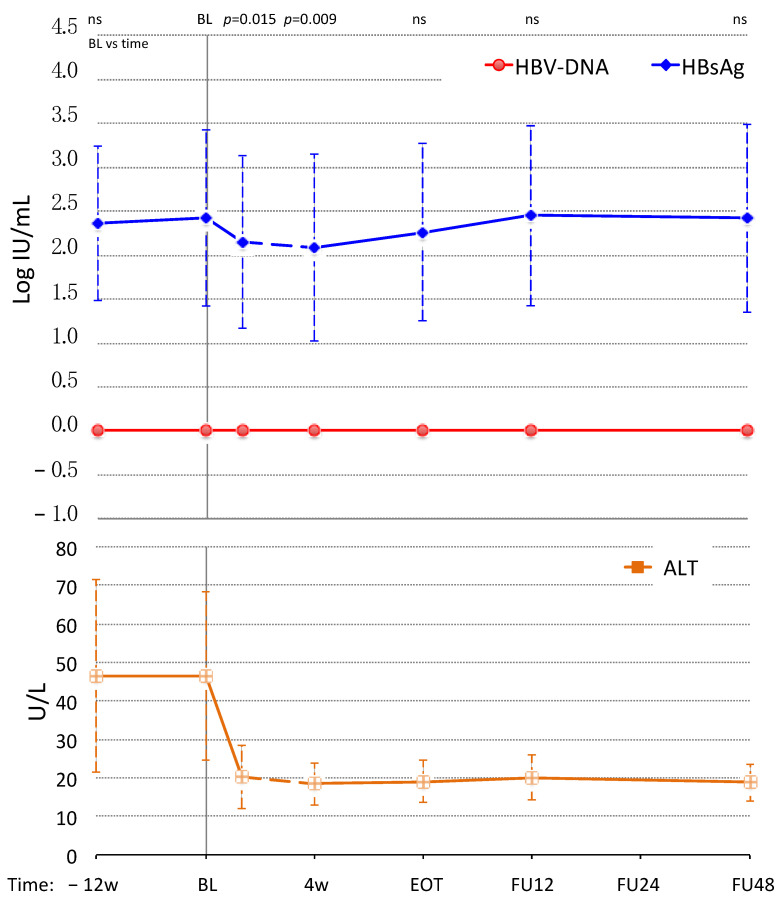
Kinetics of HBV-DNA, HBsAg and ALT across Direct Acting Antivirals (DAAs) treatment is reported for the 8 patients with concomitant HBeAg Negative Chronic Hepatitis B (CHB). All these patients were on Nucleoside Analogs (NAs) therapy before starting DAAs and continued treatment; HBV-DNA remained undetectable in all time points. Data are represented as mean values (points) with Standard Deviation (dashed lines). The baseline (BL) values were compared to the other time points (−12 w: 12 weeks before BL; 4w: week 4 of DAAs therapy; EOT: End of Therapy; FU12, FU24 and FU48: Follow Up week 12, 24 and 48) using the Student’s *t*-test for means of paired data (*p* values of the most relevant comparisons for HBsAg changes are shown on the top of the graph; ns: not significant).

**Figure 4 jcm-11-01406-f004:**
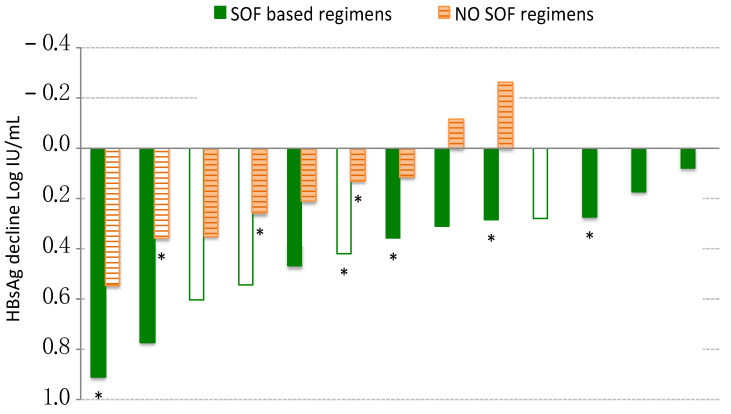
Maximum HBsAg decline observed during DAAs treatment in 22 patients. SOF: Sofosbuvir (full columns); NO SOF: No Sofosbuvir (columns with stripes). Columns without color fill represent patients of female gender. * indicates CHB patients on NAs treatment.

**Table 1 jcm-11-01406-t001:** Characteristics of the 23 patients enrolled before HCV treatment.

Parameter	Units	Overall	ENI	CHB (NAs)	*p* Value
Subjects	Number	23	15	8	
Gender	Males	17 (73.9%)	11 (73.3%)	6 (75%)	1.00
Age	years	57.1 (36.1–81.7)	62.1 (41.0–81.7)	54.8 (36.1–65.9)	0.093
LS	kPa	10.5 (5.3–38.5)	10.5 (5.3–38.5)	15.7 (6.6–35.0)	0.495
ALT	U/L	45 (9–161)	51 (9–161)	44 (16–96)	0.495
HCV-RNA	Log IU/mL	6.2 (3.7–7.7)	6.2 (3.7-7.7)	6.0 (4.9–6.7)	0.561
Genotype	1a	4 (17.4%)	2 (13.3%)	2 (25.0%)	nd
	1b	7 (30.4%)	5 (33.3%)	2 (25.0%)	nd
	2	3 (13.0%)	3 (20.0%)	0	nd
	3	7 (30.4%)	5 (33.3%)	2 (25.0%)	nd
	4	2 (8.7%)	0	2 (25.0%)	nd
HBsAg	IU/mL	34.37 (0.10–17,915.0)	6.61 (0.10–17,915.0)	213.19 (16.88–4165.7)	0.028
HBV-DNA	Detected *	7 (30.4%)	7 (46.7%)	0	nd
HBV-DNA	IU/mL **	31 (10–599)	31 (10–599)	n.a.	nd
Genotype	B	1 (4.3%)	0	1 (12.5%)	nd
	D	8 (34.8%)	4 (26.7%)	4 (50.0%)	nd
	n.a.	14 (60.9%)	11 (73.3%)	3 (37.5%)	nd

Data are reported as number (%) or median values (range), as appropriate. ENI: HBeAg Negative Infection; CHB (NAs): Chronic Hepatitis B treated with Nucleos(t)ide Analogues; LS: Liver Stiffness. ALT: Alanine-aminotransferase; nd: not done. * Detection limit for HBV-DNA = 6 IU/mL. ** In 7 patients with measured HBV-DNA (≥6 IU/mL). n.a.: not applicable.

**Table 2 jcm-11-01406-t002:** Factors influencing maximum HBsAg decline during DAAs.

Category		Log HBsAg Maximum Decline Median (Range)	Mann–Whitney U Test *p* Value	Log HBsAg >0.4 Decline Number (%)	Chi Square *p* Value
Patients	Overall	0.30 (−0.26–0.91)		7/22 (31.8)	
Gender	Males	0.27 (−0.26–0.91)	0.042	3/16 (18.7)	0.032
	Females	0.48 (0.27–0.60)		4/6 (66.7)	
HBV phase	ENI	0.29 (−0.26–0.77)	0.610	5/14 (66.7)	0.604
	CHB	0.32 (0.13–0.91)		2/8 (35.7)	
Fibrosis	F0–F2	0.21 (−0.26–0.42)	0.105	1/7 (14.3)	0.228
	F3–F4	0.35 (−0.11–0.91)		6/15 (20.0)	
DAAs	SOF	0.36 (0.08–0.91)	0.045	6/13 (46.2%)	0.083
	NO SOF	0.22 (−0.26–0.55)		1/9 (11.1%)	

Data are reported as number (%) or median values (range), as appropriate. ENI: HBeAg Negative Infection; CHB: Chronic Hepatitis B; DAAs: Direct Acting Antivirals. SOF: Sofosbuvir-based therapy; NO SOF: No Sofosbuvir based therapy.

**Table 3 jcm-11-01406-t003:** IP-10 changes during DAAs and correlations in 7 ENI and 5 CHB patients.

Parameter		BL	Week 4	EOT	EOF
HBsAg	Log IU/mL	2.09 ± 1.56	1.82 ± 1.54	1.76 ± 1.70	1.84 ± 1.68
BL vs time	*p* value		<0.001	0.002	0.062
IP-10	Log pg/mL	2.35 ± 0.30	1.66 ± 0.57	1.75 ± 0.49	1.69 ± 0.55
BL vs time	*p* value		<0.001	<0.001	0.003
IP-10 ENI	Log pg/mL	2.46 ± 0.34	1.84 ± 0.51	1.82 ± 0.56	1.76 ± 0.55
IP-10 CHB	Log pg/mL	2.22 ± 0.22	1.45 ± 0.61	1.68 ± 0.45	1.63 ± 0.61
ENI vs CHB	*p* value	0.221	0.284	0.159	0.729

Data are reported as mean ± SD. Student’s *t*-test was used to compare means of the paired data and generate the *p* values. ENI: HBeAg Negative Infection; CHB: Chronic Hepatitis B; DAAs: Direct Acting Antivirals.

## Data Availability

Data supporting reported results can be provided upon motivated request to the corresponding Authors.

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
