# Peer review of "Different Kinetics of HBV-DNA and HBsAg in HCV Coinfected Patients during DAAs Therapy"

_jcm, 2022, doi:10.3390/jcm11051406_

Round 1
Reviewer 1 Report
Your title is misleading and sounds like a review article, not a primary research paper. Be more direct with your results
51: “also extending the target patient” rephrase
80: materials and methods reads fine, but having subheadings for specific items would be helpful (Patients, Treatment Regimen, Liver Assays, PCR).
104: this section needs expansion. Potential additions: who conducted the tests, where were they conducted, which Cobas test did you use (I know it was quantitative not qualitative but you should say that; https://diagnostics.roche.com/global/en/products/params/cobas-ampliprep-cobas-taqman-hcv-test-v2-0-qualitative-and-quantitative.html), which liver tests were performed, define infectious unit, the NATs and the Ag test are completely different analysis systems
152: you should reference figure 1 in the text not as a separate sentence on 161
Figure 1: your figures need to be consistent with the text regarding punctuation, so either commas or decimal points for both. Convention would be to indicate what the vertical line at timepoint 0 represents. Figure is a little hard to read, individual plot points with error bars would drive your point home better, you should list colors in the figure legend, changing line styles would help colorblind readers, there were plenty of significant p values in your text but that’s not displayed in the figure. Same goes for figure 3
168: list the statistics in the text
Figure 2: this needs to be fixed. Excel is fine but there’s formatting problems, try exporting as a .png
190: computable?
Table 2: category is spelled wrong. You should probably list how you’re generating these statistics here, I get you said it in the methods but it can’t hurt to describe how you’re generating these p values
205: this is an inadequate paragraph. Don’t make the reader do all the work
Figure 4: formatting is really bad and makes this unreadable
226: this study
260-261: reword this sentence
305: reword “neither”
311: investigated
Reviewer 2 Report
Brief summary
23 HCV and HBV dual infected patients received DAA therapy. Elevation of HBV DNA with decrease of HBsAg and IP-10 were found during therapy in 18 NA untreated patients. Such response is temporally, although significant, did not modify individual pre-treatment profiles.
- The observation is interesting; however, the case number is quite small.
- In the materials and methods section, HBV DNA level is persistently 2000 IU/ml in HBeAg negative infection. This is confusing, higher, or lower?
- The investigators suggest ALT level declined during therapy. Since HBV DNA and HBsAg levels are associated with liver biochemistry. Please show AST and bilirubin level in Figure 1 and 3.
- In Table 2, is it possible to include maximal bilirubin level in the analysis?
- The reason of increase of HBV-DNA inversely correlated with the decline of HBsAg is difficult to analyze in this small series. If there was an increase of bilirubin level during therapy, it could be a temporally suppression of HBsAg production due to DAA liver toxicity.
Reviewer 3 Report
1. The patient number was scarce.
2. Figures were not clear enough for reading and some part was blurred.
Round 2
Reviewer 2 Report
accept
Reviewer 3 Report
No comment